# Hematological Response of Juvenile Cobia to Three Anesthetics

**Karl Sorensen [1], Steven R. Craig [2], Avner Cnaani [3] and Ewen McLean [4],***

1  Accenture, San Francisco, CA 94105, USA
2  Optimal Aquafeed, Omaha, NE 68127, USA
3  Institute of Animal Science, Agricultural Research Organization, Bet Dagan 50250, Israel
4  Aqua Cognoscenti, West Columbia, SC 29170, USA
*  Correspondence: ewen.mclean@gmail.com; Tel.:+1-(803)-463-9362

**Abstract:** Optimal concentrations of three anesthetics for use with juvenile cobia (Rachycentron canadum) were determined using time-to-recovery and hematological datasets. Buffered MS-222, clove oil and 2-phenoxyethanol (2-PE) were examined. Juvenile cobia were exposed to three concentrations of each anesthetic at 24 °C. Based on time to recovery, optimal doses for MS-222 was determined as 120-mg $L^{-1}$, that for 2-PE, 0.4-mL $L^{-1}$ and, for clove oil, 5-mL $L^{-1}$. The hematological response of cobia to anesthesia included quantification of whole blood pH, $pCO_2$, $pO_2$, and $Ca^{2+}$, $K^+$, $Na^+$, $Cl^-$, hematocrit and glucose. Irrespective of anesthetic employed, cobia expressed metabolic acidosis, with changes in blood pH ($p < 0.001$) being matched by increases ($p < 0.001$) in $pCO_2$. Anesthesia tended to increase blood $Na^+$, hematocrit, $pO_2$, $Ca^{2+}$, and $K^+$ although differential responses between anesthetics were recorded, suggesting different modes of action. A combination of recovery and hematological data indicated that when anesthesia is necessary, MS-222 represents the sedative of choice.

**Keywords:** 2-phenoxyethanol; clove oil; glucose; hyperkalaemia; MS-222; *Rachycentron canadum*

## 1. Introduction

During production and experimentation fish may be exposed to a variety of stressors [1,2] and, to avoid injury and stress during manipulation, anesthesia is often used [3,4]. In fish it is manifest that anesthetics initiate a diverse range of reactions, some of which may be considered detrimental. These include alterations to normal hematological and hormonal profiles, changes to the cardiovascular system, negative impacts on gametes, disease resistance and immunity, and damage to olfactory and other sensory processing systems. Evident also is that reactions to anesthesia are species, size/age, anesthetic, dose, and temperature dependent [5–8]. Because of these variations it is clearly prudent to determine optimal doses for different anesthetics on a species-by-species basis while, from a research perspective, it is vital that the effects of different anesthetics on fish hematological parameters are established since these may impact the interpretation of experimental results [9].

While not having achieved its anticipated levels of global production [10], cobia *Rachycentron canadum*, still represents an important species with >40,000 tonnes being farmed annually since 2010, mainly in China, Taiwan, Vietnam, and Panama [11]. The weak growth in uptake of cobia aquaculture has not reduced interest in this species, however, since it expresses many favorable attributes including, among others, rapid growth, acceptance of alternative proteins, hardiness, resistance to disease and flesh quality [12,13]. Research with cobia progresses at pace and there is a need to establish the precise impact of commercial and research sedatives upon their overall physiology. Indeed, several studies have already examined the response of cobia at different ages and under varying conditions of husbandry to anesthesia [14–17]. However, the effect of various anesthetics on blood gas and ion dynamics of cobia remains unknown. In order to fill this knowledge gap,

here we examine the dose response of cobia to three anesthetics, *viz.* 3-aminobenzoic acid ethyl ester methanesulfonate (MS-222), 2-phenoxyethanol (2-PE), and clove oil, and, using optimal doses of each, examine their influence on various hematological parameters in juvenile cobia.

## 2. Materials and Methods

### 2.1. Animals, System and Husbandry

Cobias were acclimated for a period of 3 months in two 2.5 m$^3$ tanks set in a recirculating life support system mode comprising: a 750 L KMT-based fluidized bed biofilter (Kaldnes Miljøteknologi, Tønsberg, Norway), a bead filter for solids removal (Aquaculture Technologies Inc., Metaire, LA, USA), a protein skimmer (Aquaculture Technologies Inc., Metarie, LA, USA), and two 40-watt UV sterilizers (Aquatic Ecosystems, Apopka, FL, USA). The fluidized bed was oxygenated using diffusion air lines connected to a 1 hp Sweetwater remote drive regenerative blower (Aquatic Ecosystems, Apopka, FL, USA). During acclimation, water temperature (21.5 ± 1.5 °C) and pH (7.80 ± 0.26) were monitored 3 times a week using a Hanna Instrument 9024 pH meter (Aquatic Ecosystems, Apopka, FL, USA). Dissolved oxygen (5.63 ± 0.77 mg L$^{-1}$) and total ammonia nitrogen (0.42 ± 0.21 mg L$^{-1}$) were also measured three times a week using an YSI 85 Series dissolved oxygen meter (YSI Inc., Yellow Springs, OH, USA) and by spectrophotometric analysis (Hach Inc., Loveland, CO, USA), respectively. Nitrite (0.61 ± 0.50 mg L$^{-1}$) and nitrate (62 ± 9 mg L$^{-1}$) levels were quantified once a week by spectrophotometric analysis (Hach). Salinity was monitored and maintained near 19‰ (19.2 ± 1.8‰) using Crystal Sea synthetic sea salt (Marineland, Baltimore, MD, USA). Throughout acclimation and following anesthesia, fish were hand fed twice daily to satiation with a 3 mm extruded diet (50% protein, 20% lipid; Corey Feed Mills Ltd., Fredericton, NB, Canada). Animals were kept on a 12:12 photophase-scotophase light cycle with 30 min dawn-dusk dimming.

### 2.2. Anesthesia, Sample Collection and Data Analysis

#### 2.2.1. Dose Response

Experiments were undertaken using cobia of 66.7 ± 12.2 g and 221 ± 10.90 mm TL. For each trial, individual animals were randomly taken from the holding tank and placed into a 25 L bucket containing different doses of oxygenated anesthetic in a volume of 10 L water taken from the holding tank. The doses of anesthetic used were clove oil (Sigma-Aldrich, St. Louis, MO, USA): 5, 10 and 15 mL L$^{-1}$; 2-phenoxyethanol (2-PE; Sigma-Aldrich): 0.3, 0.4 and 0.5 mL/L, and ethyl 3-aminobenzoate methane sulfonate (MS-222; Sigma-Aldrich): 80, 100 and 120 mg L$^{-1}$. Clove oil and 2-PE were pre-dissolved in ethanol as previously described [18]. MS-222 was buffered using sodium bicarbonate as recommended by Cotter and Rodruck [19]. The pH of anesthetic baths was 7.77 ± 0.29. At the end of individual observations each anesthetic bath and recovery water were replaced with freshly prepared solutions. This was repeated 5 times for each dose of anesthetic (i.e., *n* = 5 fish per dose examined). Entry into and recovery from anesthesia were monitored according to the stages described in Table 1 [20]; a stopwatch was used. Once fish attained stage 5 (loss of reflex reactivity), they were immediately removed from the anesthetic bath and transferred to a recovery vessel containing water derived from the recirculating life support system. The recovery vessel water was supplied with pure oxygen. Total recovery (stage 5) was considered complete when fish expressed normal swimming and reflex behavior. Following revival, fish were placed into a holding tank and their survival and behavior recorded over a 5 d period.

#### 2.2.2. Blood Collection, Handling, and Analysis

The hematological response of cobia to anesthesia was examined using the optimum dose for each chemical (*n* = 10 per dose). This was considered as the concentration from which experimental fish recovered most rapidly. Where no differences were observed in recovery times between doses, the lowest concentration was used. For clove oil the

optimum dose was considered as 5 mL $L^{-1}$, for 2-PE 0.4 mL $L^{-1}$ and for MS-222 120 mg $L^{-1}$. The identical procedures of anesthesia described previously were repeated and following full recovery (stage 5) blood was rapidly collected from individual fish into 1 mL pre-heparinized tuberculin syringes. The sample taken was mixed arterial-venous blood from the caudal artery-vein complex. Immediately following collection, blood samples were aspirated into an ABL80 FLEX Analyzer (Radiometer, Copenhagen, Denmark) and the following parameters quantified: hematocrit, pH, $pCO_2$, $pO_2$, and $Ca^{2+}$, $K^+$, $Na^+$ and $Cl^-$. To obtain a relative difference measurement for blood glucose response [21] a MediSense Precision Blood Glucose Monitor (Abbott Laboratories, Lake County, IL, USA) was used. Bled fish were returned to their holding tank and their survival and behavior recorded over a 5-d period.

**Table 1.** Behavioral alterations of cobia to various stages of anesthesia and recovery therefrom, modified from Keene et al. [20].

| Sedation | | |
|---|---|---|
| *Stage* | *Descriptor* | *Behavior* |
| 0 | Normal equilibrium | Reactive to external stimuli with opercular rate and muscle tone normal. |
| 1 | Light sedation | Slight loss of reactivity to external visual and tactile stimuli. Opercular rate slightly decreased but equilibrium normal. |
| 2 | Deep sedation | Total loss of reactivity to external stimuli except to strong pressure. Slight decrease in opercular rate but equilibrium normal. |
| 3 | Partial decline in equilibrium | Partial loss of muscle tone combined with erratic swimming. Increased opercular rate. Only reactive to strong tactile and vibrational stimuli. |
| 4 | Total loss of equilibrium | Total loss of muscle tone and equilibrium. Slow but regular opercular rate combined with loss of spinal reflexes. |
| 5 | Lack of reflex reactivity | Total loss of reactivity. Opercular reactions slow and irregular. Heart rate very slow and loss of all reflexes. |
| Recovery | | |
| *Stage* | | *Behavioral adjustment* |
| 1 | | Return of opercular movement. |
| 2 | | Partial recovery of equilibrium and swimming motion. |
| 3 | | Total recovery of equilibrium. |
| 4 | | Revival of avoidance swimming motion and response to external stimuli, but behavioral response impassive. |
| 5 | | Total behavioral recovery. Normal swimming and reflex activity. |

*2.3. Statistical Analysis*

All data were analyzed by analysis of variance using the SAS 9.1 statistical program (SAS, Cary, NC, USA). When appropriate, data were also subjected to Duncan's multiple range tests for means separation. Significant differences among the data were observed when $\alpha < 0.05$.

### 3. Results

*3.1. Dose Response*

All animals subjected to anesthesia recovered fully and on return to their holding tank rapidly established normal swimming behavior. No mortalities occurred 5 days post-anesthesia, irrespective of anesthetic or dose employed. The time for fish to recover from anesthesia, regardless of type or dose, was always longer than the time taken for fish to succumb to stage 5 narcosis. The time taken for cobia to reach stage 5 anesthesia when exposed to the different concentrations of each of the three anesthetics, and the recovery time, are presented in Figure 1. Briefly, time to reach stage 5 anesthesia in fish exposed to

clove oil did not differ significantly with dose. The lowest recorded time was observed when fish were immersed in 10 mL L$^{-1}$ clove oil, followed by the 5 mL L$^{-1}$ concentration, and the 15 mL L$^{-1}$ concentration. For 2-PE, time to stage 5 anesthesia followed the dose–response, with cobia immersed in the highest concentration succumbing more rapidly with the order: 50-, 40-, and 30-mL L$^{-1}$ 2-PE. The 30-mL L$^{-1}$ dose induced fish to reach stage 5 more slowly ($p < 0.02$) than either the 40- or 50-mL L$^{-1}$ doses, wherein fish responded identically. In the MS-222 fish exposed to the 100-mg L$^{-1}$ dose were the fastest to reach stage 5 anesthesia and fish exposed to 120-mg L$^{-1}$ were the slowest, but the recorded differences were not significant (Figure 1).

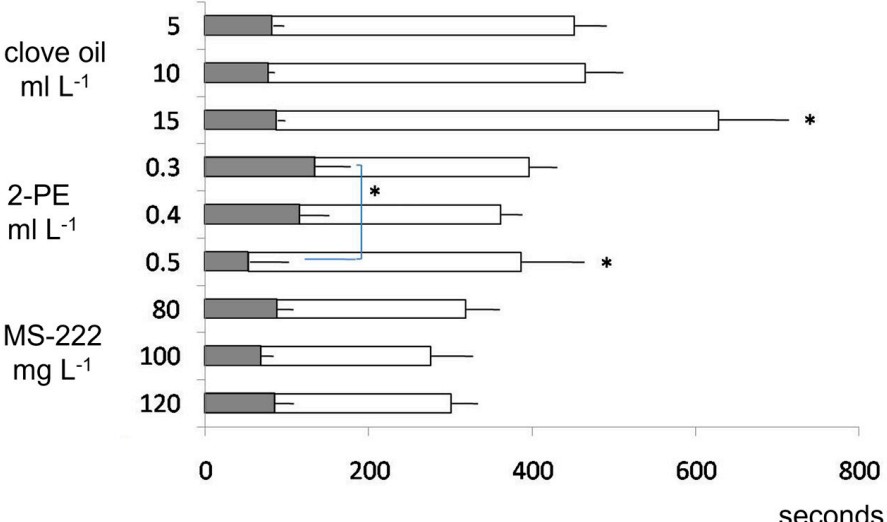

**Figure 1.** The response of juvenile cobia to three different anesthetics presented at three different doses. The gray colored series for each dose represents time to stage 5 anesthesia whereas the white series represents time to full recovery from anesthesia. Bars are ± S.D. Doses are in mL L$^{-1}$ for clove oil and 2-phenoxyethanol and mg L$^{-1}$ for MS-222. Asterisks indicate significant differences for individual anesthetics at the $\alpha = 0.02$ level.

Recovery of cobia from clove oil anesthesia differed significantly ($p < 0.004$) followed the dose–response in the order, with fish exposed to the highest concentration (15-mL L$^{-1}$) recovering slower than those exposed to the lower doses (5- and 10-mL L$^{-1}$). Recovery from the 30- and 40-mL L$^{-1}$ of 2-phenoxyethanol was significantly ($p < 0.0005$) faster than the 50-mL L$^{-1}$ dose. No differences were observed in the times taken for fish to recover from MS-222 irrespective of dose employed (Figure 1).

### 3.2. Effect of Anesthesia on Hematological Parameters

The effect of optimum doses of the different anesthetics on various blood parameters is summarized in Table 2. Anesthesia had variable impacts on the hematology of exposed fish. A universal response observed following anesthesia, however, was a decrease in blood pH and increase in pCO$_2$ ($p < 0.001$). When compared to controls, both MS-222 and clove oil expressed elevated ($p < 0.001$) blood Na$^+$ and Ca$^{2+}$ levels. Highest concentrations for both Na$^+$ and Ca$^{2+}$ were recorded in fish exposed to clove oil. Blood glucose, pO$_2$, K$^+$ and hematocrit levels were higher in fish anesthetized with 2-PE when compared against control cobia. However, no differences in blood glucose or K$^+$ were discerned between control, clove oil and MS-222 treated fish. Differences ($p < 0.005$) were observed for blood Na$^+$ and pCO$_2$ levels between clove oil and 2-PE (Table 2), whereas blood chloride levels (157.3 ± 7.4 mmol L$^{-1}$) remained unaffected by treatment ($p > 0.05$).

**Table 2.** The effect of clove oil (5 mL L$^{-1}$), 2-phenoxyethanol (2-PE; 0.4 mL L$^{-1}$) and MS-222 (120 mg L$^{-1}$) on blood pH, hematocrit electrolyte levels, glucose and pCO$_2$ and pO$_2$ in juvenile cobia. Different superscripts in individual columns signify significant difference at the $\alpha < 0.05$ level by Duncan's multiple range test.

| | Hematocrit | pH | $p$CO$_2$ | $p$O$_2$ |
|---|---|---|---|---|
| **Control** | 23.2 ± 4.0 | 7.58 ± 0.04 [a] | 9.0 ± 1.23 [c] | 30.2 ± 9.5 |
| **Clove oil** | 29.2 ± 5.3 | 7.27 ± 0.08 [b] | 15.8 ± 1.48 [a] | 41.6 ± 13.5 |
| **2-PE** | 30.8 ± 4.4 | 7.23 ± 0.06 [b] | 13.0 ± 1.22 [b] | 59.0 ± 21.1 |
| **MS-222** | 25.8 ± 2.6 | 7.28 ± 0.03 [b] | 14.4 ± 1.34 [ab] | 51.6 ± 19.0 |
| *p-value* | 0.0568 | 0.001 | 0.001 | 0.068 |
| | **Na$^+$ (mmol L$^{-1}$)** | **K$^+$ (mmol L$^{-1}$)** | **Ca$^{2+}$ (mmol L$^{-1}$)** | **Glucose (mg dL$^{-1}$)** |
| **Control** | 165.8 ± 1.92 [c] | 4.65 ± 0.41 [b] | 1.52 ± 0.01 [b] | 56.2 ± 6.61 [b] |
| **Clove oil** | 175.8 ± 4.49 [a] | 5.44 ± 0.72 [ab] | 1.76 ± 0.17 [a] | 54.6 ± 11.74 [ab] |
| **2-PE** | 170.4 ± 4.04 [bc] | 5.94 ± 0.56 [a] | 1.64 ± 0.12 [ab] | 83.8 ± 20.9 [a] |
| **MS-222** | 172.2 ± 3.96 [ab] | 5.38 ± 0.55 [ab] | 1.70 ± 0.14 [a] | 74.8 ± 16.42 [ab] |
| *p-value* | 0.005 | 0.036 | 0.052 | 0.017 |

## 4. Discussion

During intensive aquaculture anesthesia may be used during spawning, transportation, vaccination and for grading purposes. Often, for research compliance, there is a mandated use of anesthetics, and it is generally accepted, as demonstrated by the studies herein with cobia, that the physiological reaction of fish to sedatives varies with anesthetic and the dose and duration of exposure used [3,8]. The speed at which cobia succumbed to anesthesia was generally more rapid than recorded for other teleosts [22,23], which may reflect different test temperatures and fish sizes studied or represent a species distinction. Other experiments with cobia, using similar doses of anesthetic, report comparable times to induction for the three anesthetics examined here [15,16]. Times to recovery, however, while similar for MS-222 and 2-PE, were protracted for clove oil. This difference likely occurred due to differences between distilled eugenol and clove oil which, in addition to eugenol, comprises ß-caryophyllene, α-humulene, eugenyl acetate, and trace amounts of other compounds [24,25]. Larger fish and higher temperatures may also have had an influence. The generally lower induction time recorded with clove oil may be related to its high lipid solubility and ability to penetrate cell membranes [20] whereas the extended recovery period, relative to 2-PE and MS-222 treated fish, may have resulted due to the oil blocking gaseous exchange across the gill epithelia [26].

Using time to recover from anesthesia, a dose of 5 mL clove oil L$^{-1}$ was considered optimum while, based on total time for induction through complete recovery, the optimum dose of 2-PE for minimal physical manipulations of cobia was considered 0.4 mL L$^{-1}$. A clear dose-dependent response of fish to MS-222 anesthesia has been reported for many species and recommended doses of MS-222 for teleosts range between 50 and 480 mg L$^{-1}$ [3,4,8]. In cobia, however, no differences were recorded in the times taken for fish to succumb to or recover from MS-222 anesthesia at the 3 three doses used. The reasons for this contradiction remain obscure but it is possible that the lowest dose of MS-222 employed exerted a maximal physiological effect. Further studies are warranted to establish a more precise dose response for cobia and MS-222 and the physiological impact of this anesthetic on the animal. Nevertheless, in general, cobia in the current study declined into and recovered from MS-222 anesthesia more rapidly than observed for other test anesthetics. The more rapid recovery times for MS-222 may be related to the initial stimulatory effect that it has upon the heart rate [27] in contrast to clove oil, which has an inhibitory effect [28]. Moreover, other than a mild metabolic acidosis and disturbance in blood calcium levels, the physiological response of cobia to MS-222, when compared to clove oil and 2-PE,

was marginal such that, of the three test compounds examined, MS-222 represented the anesthetic of choice. The dose of 120 mg L$^{-1}$ was selected as optimal due to the lower variation experienced in recovery times.

A common response of fish to anesthesia is hemoconcentration [29,30] and cobia expressed elevated hematocrit levels after narcosis; especially when exposed to 2-PE. Anesthesia reduces ventilation rates and thus causes a hypoxic response through asphyxia [31]. Fish compensate for asphyxiation by increasing red blood cell numbers, either through splenic contraction [32] and or by heightened erythropoiesis [33]. Another mechanism that may be employed to counteract hypoxia is a reduction in cardiovascular resistance which would permit the heart to pump larger volumes of blood, albeit at reduced stroke rates. Support for this process has been elegantly provided by Rothwell and Forster [34] who illustrated that eugenol and MS-222 decrease tail blood vessel tension in vitro. Declining ventilation rates, together with physical exertion during capture, causes metabolic acidosis and the reductions in cobia blood pH may have occurred due to a buildup of lactate during capture [30]. Associated with acidosis and irrespective of anesthetic, were elevations in $p$CO$_2$. Under normal circumstances, high $p$CO$_2$ decreases the affinity of hemoglobin for oxygen [35], causing increased dissociation of O$_2$ from hemoglobin thereby facilitating O$_2$ transport into tissues. This response generally results in a lowered $p$O$_2$ [36] but in the present study, cobia $p$O$_2$ levels remained stable. This inconsistency was probably the result of using pure oxygen to aerate the anesthetic recovery baths.

Only 2-PE caused a significant rise in glucose when compared to control fish. One explanation for this hyperglycemic reaction might be a more rapid onset of the stress response to this chemical. Alternatively, hyperglycemia may result due to hypoxic conditions at the cellular level, induced by anesthetics. Under such conditions there would be a shift toward anaerobic metabolism and higher demand for glucose to meet physiological requirements using a less energetically efficient pathway. Alternatively, the speed of the glucose response may be protracted in MS-222 and clove oil treated fish, since anesthesia increased blood glucose between 30- and 60-min post-exposure to MS-222 and eugenol, respectively [15].

All anesthetics examined induced acidosis, in addition to increased blood Na$^+$ and K$^+$ levels but were without effect on Cl$^-$. These observations may be explained as acid-base regulation, due to the increased $p$CO$_2$ levels. Hypercapnia affects blood pH, inducing acidosis, which in turn increases proton excretion across the gills. Although sodium influx and proton efflux are not linked by a Na$^+$/H$^+$ (NH$_4^+$) antiport, they are electrically coupled. The negative potential of the inner side of the membrane generated by proton transport is the driving force for sodium uptake [27]. Transport of Na$^+$ from gill epithelium into the blood, and of H$^+$ from the blood into the gill epithelium is mediated by Na$^+$/K$^+$(NH$_4^+$)-ATPase [37], and the observed hyperkalaemia may be due to release of stored potassium needed for this ion exchange process.

In conclusion, examination of blood ions, pH and glucose illustrate subtle variations in the action for 2-PE when compared to MS-222 and clove oil, with 2-PE expressing a stronger physiological effect. Additionally, because recovery from clove oil was highly extended, we suggest that when sedation is essential for cobia, MS-222 represents the anesthetic of choice.

**Author Contributions:** K.S.: Data Curation, Formal Analysis, Investigation. S.R.C.: Formal Analysis, Investigation, Visualization, Writing—Review and editing. A.C.: Investigation, Writing—Review and editing. E.M.: Conceptualization, Methodology, Writing—original draft. All authors have read and agreed to the published version of the manuscript.

**Funding:** This research received no external funding.

**Institutional Review Board Statement:** No animals were sacrificed during the present study and fish were anesthetized during blood withdrawal with appropriate regard to Virginia Tech's Institutional Animal Care and Use Committee, who approved submitted research protocols. All described research complied with all relevant internal and international animal welfare laws, guidelines, and policies.

**Data Availability Statement:** The data are available upon request from the authors.

**Conflicts of Interest:** The authors declare no conflict of interest.

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
