# Peer review of "Hematological Response of Juvenile Cobia to Three Anesthetics"

_fishes, doi:10.3390/fishes8010031_

Round 1

Reviewer 1 Report

 According to the length and knowledge presented in the MS of

 Sorensen et al. could be considered as a short communication, not a full-length article.

The dose response study must be include statistical dose response analysis such as polynomial contrast and fit regression models. Which, is absent in the current investigation. Accordingly, the title and other related sections need to be corrected.

The keywords need to reorder alphabetically.

L178 this sentence is not clearly presented "Different figures in parameter columns signify significant difference…."

Figure 1 is in low quality, it needs to be improved (color and higher resolution figure is recommended)

In a separate table mentioned all stages of anesthesia and recovery according to the cited reference.

In table 1 footnote, the evaluated doses have to mention, where each table should be stand-alone.

L193-194: update the references and use the right ref. style (Randall, 1962) and (Hikasa et al., 1986).

For ms 222 the recommended dose is 120 however no differences with lower levels so the recommended level should be the lower dose to reduce the use of chemicals and improve cost-efficient.

L 215: This information is inconsistence with M and M line 59 {This inconsistency was probably the result of using pure oxygen to aerate the anesthetic recovery baths} and [connected diffusion air lines connected; this in the filter], So revised this part.

The conclusion section is a part of the discussion, it needs to be rewritten based on the obtained results and the final recommendation. However, lines 237-248 could be merged in the discussion

Author Response

As suggested by the reviewer, the title of the manuscript has been modified to take account of data analyses employed.

Key words have been listed alphabetically.

The legend to Table1 (new Table 2) has been modified to clarify content.

Figure 1 has been chamged in accordance with the reviewer's request.

An additional Table (new Table 1) has been incorporated into the text as recommend by the reviewer to delineate different stages of anesthesia and recovery.

The doses of anesthetics employed to examine hematological response have, as recommended by the reviewer, been incorporated into the Table legend.

L 193-194: the citation method has been adjusted to journal style as indicated by the reviwer.

As noted in the discussion, the recommended dose of MS-222 (120 mg/L rather than 100 mg/L) was selected based upon lower variation in recovery times for sedated fish. 

Previous L 215 has been revised through modification of M&M section.

As recommended by the reviewer, the conclusion has beend deleted and directly appended into the discussion section.

Reviewer 2 Report

Sorensen et al. reported an interesting study on the effect of three different main anaesthetic drugs on haematological values in juvenile cobia. Data are well-collected and documented.

I suggest the acceptance after minor revisions.

Minor Revision: 

1) I recommend specifying the specie of cobia

2) The dose of each anaesthetic used in table 1 for the evaluation of haematological values must be specified in the main text. 

3) Why did the authors not analyse the haematological values for each dose of anaesthetic drugs? Please, comment on it.

4) With equal times to reach stage 5 sleep and recovery times, why didn't the authors evaluate which dose of each drug had a minor impact on blood values? Please, comment on it.

Author Response

The genuis and species of cobia has been incorporated into the key word section and following first use in  the text.

Adjustmenmts made with respect to reviewer 1 comments take care of all other recommendations.

Round 2

Reviewer 1 Report

·         Thanks for response but next time the authors have to repley for all points as “question and reply”.

 ·         Im still also suggesting considering the study of Sorensen et al. as a short communication, not a full-length article depend on the length and knowledge presented in the MS.

 ·         In the caption of fig. 1 I suggest removing “left” or “right” words to indicate the column to remove any potential confusion, the color is enough, also the column is overlapped.

 ·         May the authors misunderstanding me about the conclusion, the conclusion must be written based on the obtained results not repeat the discussion and references, so delete the references in the conclusion and give us take home message.

Author Response

Q: Im still also suggesting considering the study of Sorensen et al. as a short communication, not a full-length article depend on the length and knowledge presented in the MS.

R: The authors have no issue with this suggestion but can find no instructions in the Author Instructions documentation for preparation of short communications and thus are compelled to leave the manuscript essentially unchanged. 

Q: In the caption of fig. 1 I suggest removing “left” or “right” words to indicate the column to remove any potential confusion, the color is enough, also the column is overlapped.

R: We have removed the "left" and "right" words from the legend as requested. 

Q: May the authors misunderstanding me about the conclusion, the conclusion must be written based on the obtained results not repeat the discussion and references, so delete the references in the conclusion and give us take home message.

R: We have reduced the final paragraph by ~60% to highlight the takehome message/conclusion. Appropriate references have been removed which may require rearrangement of reference list..